# CA-IX-Expressing Small Extracellular Vesicles (sEVs) Are Released by Melanoma Cells under Hypoxia and in the Blood of Advanced Melanoma Patients

**DOI:** 10.3390/ijms24076122

**Published:** 2023-03-24

**Authors:** Marta Venturella, Alessandro Falsini, Federica Coppola, Gaia Giuntini, Fabio Carraro, Davide Zocco, Antonio Chiesi, Antonella Naldini

**Affiliations:** 1Cellular and Molecular Physiology Unit, Department of Molecular and Developmental Medicine, University of Siena, Via A. Moro 2, 53100 Siena, Italy; marta.venturella@student.unisi.it (M.V.); alessandro.falsini@student.unisi.it (A.F.); federica.coppola@student.unisi.it (F.C.);; 2Cellular and Molecular Physiology Unit, Department of Medical Biotechnologies, University of Siena, Via A. Moro 2, 53100 Siena, Italy; fabio.carraro@unisi.it; 3Lonza Siena, Strada del Petriccio e Belriguardo 35, 53100 Siena, Italy; davide.zocco@lonza.com; 4Exosomics SpA, Strada del Petriccio e Belriguardo 35, 53100 Siena, Italy; achiesi@exosomics.eu

**Keywords:** melanoma, sEVs, hypoxia, HIF-1, carbonic anhydrase IX, biomarker discovery, liquid biopsy

## Abstract

Cutaneous melanoma is a highly aggressive skin cancer, with poor prognosis. The tumor microenvironment is characterized by areas of hypoxia. Carbonic anhydrase IX (CA-IX) is a marker of tumor hypoxia and its expression is regulated by hypoxia-inducible factor-1 (HIF-1). CA-IX has been found to be highly expressed in invasive melanomas. In this study, we investigated the effects of hypoxia on the release of small extracellular vesicles (sEVs) in two melanoma in vitro models. We demonstrated that melanoma cells release sEVs under both normoxic and hypoxic conditions, but only hypoxia-induced sEVs express CA-IX mRNA and protein. Moreover, we optimized an ELISA assay to provide evidence for CA-IX protein expression on the membranes of the sEVs. These CA-IX-positive sEVs may be exploited as potential biomarkers for liquid biopsy.

## 1. Introduction

### 1.1. Small Extracellular Vesicles

Small extracellular vesicles (sEVs) are a subtype of extracellular vesicles with sizes ranging from 30 nm to 150 nm. They originate from intracellular multivesicular bodies (MVB) that fuse with the cell plasma membrane, releasing their content, the sEVs, in the extracellular microenvironment [1,2]. EV membranes are enriched in cholesterol, sphingomyelin, ceramide, and lipid raft associated proteins [3]. Protein constituents of sEVs are tetraspanins (CD9, CD63, CD81), proteins involved in MVB biogenesis such as ALG2-interacting protein X (ALIX) and tumor susceptibility gene 101 protein (TSG101), heat shock proteins (Hsp90 and Hsc70), and transport proteins (GTPases, annexins and flotillin) [3,4,5].

SEVs are secreted by all cell types including cancer cells [4,5,6,7]. They mediate cell-to-cell communication under physiological and pathological conditions through the transfer of specific cargo. EV cargo is characterized by many species of RNA (microRNA, messenger RNA, small non coding RNA, ribosomal RNA and transfer RNA), several types of DNA (mitochondrial DNA, single-stranded DNA, and double stranded DNA) and proteins.

Tumor cells actively release sEVs to transfer information to other malignant or normal cells, both within the tumor microenvironment as well as systemically to distant tissues. It has been widely demonstrated that tumor-derived sEVs have a role in cancer progression, specific to cancer type, genetics, and stage [8,9,10].

SEVs are stable in circulation and can be isolated from all body fluids (blood, urine, saliva, milk etc.) as well as from cell conditioned media. They protect biomolecules, potential biomarkers, from degradation. Thus, sEVs are considered as candidates for liquid biopsy in the case of the absence of tissue, for early diagnoses and even for therapy efficacy monitoring [8,11,12,13,14,15,16,17,18].

### 1.2. Melanoma

Melanoma is one of the most aggressive types of skin cancer, which arises from transformed melanocytes, and despite constituting 1% of skin tumors, it is responsible for over 80% of skin cancer deaths [19]. The World Health Organization classifies melanocytic alterations into nine different categories according to the associated cumulative sun damage (CSD), as UV radiation correlates with DNA damage and melanoma spread, even though genetic predisposition can also be considered [20]. Most common mutations are related to genes controlling cell proliferation and survival (BRAF, NRAS, and KIT, respectively). Molecules such as BRAF inhibitors are studied in clinical trials, but since melanoma has great abilities to migrate and invade other tissues, the first target is the chirurgical resection of the primitive lesion in order to obtain a good prognosis [21].

### 1.3. Hypoxia

Hypoxia, described as low pO_2_ tension, characterizes the tumor microenvironment and is associated with an increased ability of invasion and metastatic potential in primary melanoma [22,23]. In addition, melanoma metastasis is not vascularized in the early state and adapt to the initial hypoxic condition, which can lead to an even more aggressive phenotype [24,25,26,27]. Hypoxia activates hypoxia-inducible factor-1 (HIF-1), which promotes the transcription of several genes involved in biological mechanisms related to melanoma progression such as angiogenesis, cell proliferation, and metastasis [28,29].

### 1.4. Carbonic Anhydrase IX

Carbonic anhydrase IX (CA-IX) is a promising marker of cellular hypoxia. CA-IX expression is driven by HIF-1, which binds to a hypoxia-response element (HRE) consensus sequence localized near the transcription initiation site of the CA9 gene [30].

CA-IX belongs to the family of carbonic anhydrases (CA) and is a transmembrane zinc metalloenzyme that catalyzes the reversible hydration of carbon dioxide to bicarbonate and protons. It plays a crucial role in maintaining the cell pH homeostasis. The expression of CA-IX in normal tissues is restricted to the stomach, duodenum, small intestine, and gallbladder. Instead, CA-IX overexpression is associated with a variety of solid cancers including melanomas [31,32,33,34,35,36,37,38,39]. In cancer, CA-IX maintains a favorable intracellular pH for tumor cells, thus increasing their survival and growth, and is correlated with aggressive phenotypes, the maintenance of stemness properties, therapy resistance, and poor prognosis [33,40,41,42,43].

### 1.5. Aim of This Study

Few studies have reported the relationship between CA-IX and sEVs in different cancer types [44,45,46,47]. For the first time, we investigated CA-IX protein expression associated with sEVs in melanoma in vitro models. We further demonstrated the potential application of CA-IX-positive sEVs as biomarkers for cancer diagnosis.

## 2. Results

### 2.1. Expression of HIF-1α and Carbonic Anhydrase IX Is Induced under Hypoxia in Melanoma Cells

The A375 and SK-MEL-28 cell lines are human malignant melanoma cell lines, respectively, the most and least aggressive model [48]. They were cultivated under both a normoxic condition (20% O_2_~pO_2_ of 140 mmHg) and, for 24 h, under a hypoxic condition (2% O_2_ = pO_2_ of 14 mmHg). Cell lysates were analyzed by Western blotting in order to investigate the expression of HIF-1α, CA-IX, and a housekeeping protein (beta actin). The results are shown in Figure 1. Both cell lines expressed significantly higher protein levels of HIF-1α under the hypoxic condition, confirming the hypoxic state of cell cultures. As expected, CA-IX protein expression was also induced under hypoxia in the A375 and SK-MEL-28 cells, since its expression is driven by HIF-1α. Moreover, the levels of CA-IX expression were higher in the A375 model compared to SK-MEL-28, and this was also an expected result due to the more aggressive phenotype of the A375 cells. Expression levels of the control protein (beta actin) were similar in all of the samples.

### 2.2. Small EVs Released by Melanoma Cells under Normoxic and Hypoxic Conditions Have Comparable Particle Concentration and Median Size

First, small EVs (sEVs) released by normoxic and hypoxic melanoma cells were characterized in terms of the total particle concentration and size. Equal amounts of CCM from each condition were harvested to purify sEVs by SEC chromatography (Appendix A). Purified sEVs were analyzed at Flow NanoAnalyzer to assess the particle’s concentration and size distribution profile. The concentration and total number of particles released by the A375 cells in normoxia were found to be slightly higher than after hypoxic treatment (Figure 2A). The same trend was observed for the SK-MEL-28 cells; however, these differences did not reach statistical significance (Figure 2A).

All samples had comparable size distribution profiles within the expected size range of sEVs (Figure 2B) with median size values between 64 and 79 nm (Figure 2C).

### 2.3. Melanoma Cells Release CA-IX-Expressing sEVs during Hypoxia

Next, we investigated whether sEVs carry hypoxia-induced markers after hypoxic treatment. Carbonic anhydrase IX (CA-IX) was selected as a hypoxia-induced marker due to its overexpression in melanoma [31,37].

To determine the expression of CA-IX mRNA on sEVs, the total RNA was extracted from equal amounts of EVs produced from each condition. Digital droplet PCR was used to quantify the CA-IX and beta actin mRNA copies. Small EVs from the A375 cells were found to carry detectable levels of the CA-IX target even in normoxic condition. Under the hypoxic condition, the expression levels of CA-IX mRNA increased significantly up to 35-fold (Figure 3A, *p* value < 0.005). Conversely, CA-IX mRNA was not detected in sEVs from the SK-MEL-28 cells in both the normoxic and hypoxic conditions (Figure 3B).

To evaluate the expression of CA-IX protein, equal numbers of sEVs were lysed and analyzed by Western blotting. Consistent with the mRNA expression data, immunoblotting analysis showed the CA-IX protein expressed on sEVs from the hypoxic A375 melanoma cells. The canonical sEV marker (Alix) was found on sEVs under both normoxic and hypoxic conditions (Figure 3C). The CA-IX protein was not found on sEVs from the hypoxic SK-MEL-28 cells, while Alix was expressed in both conditions (Appendix A).

### 2.4. CA-IX Protein Is Associated to the Membrane of sEVs Secreted by Hypoxic Melanoma Cells

Next, we wanted to demonstrate that CA-IX is expressed on the membrane of sEVs after hypoxic treatment. First, we tested the ability of an anti-CA-IX antibody, kindly provided by Exosomics SpA, to bind the membrane-associated CA-IX. To do so, sEVs were purified from CCM from a CA-IX-overexpressing HEK293 cell line (HEK293-pRTS-CA9) and used as the target for a binding kinetic experiment with the selected antibody.

Figure 4A shows the concentration-dependent binding of the antibody to CA-IX-positive sEVs. Since a soluble form of CA-IX has been detected in the sera of patients with different cancer types and in the cell culture media [49,50,51,52], concentration-dependent binding of the antibody to recombinant soluble CA-IX was confirmed using the same SPR-Biacore setup (Figure 4B).

To ascertain the presence of nanoparticles expressing canonical sEV markers in our samples, an ELISA assay was performed to target the tetraspanin CD9 on sEVs from CCM from both cell lines grown in normoxic and hypoxic conditions (Figure 5A). Consistent with previous results, the amount of CD9-positive particles did not significantly differ between the samples from different cell lines and conditions (Figure 5A).

To specifically detect sEV-associated CA-IX, samples were tested by a specific ELISA assay provided by Exosomics SpA. Consistent with our previous data, only hypoxic sEVs from A375 cells resulted in a positive signal, suggesting direct binding to EV-associated CA-IX. After detergent treatment with the RIPA buffer, the positive signal from A375 hypoxic sEVs was abrogated due to the disruption of EV membranes (Figure 5B). These results indicate that CA-IX protein is expressed on the surface of A375 sEVs released under hypoxia.

### 2.5. CA-IX-Positive sEVs Are Found in Melanoma Liquid Biopsies and Their Capture Enables the Enrichment of BRAFV600E Mutation

To investigate the release of CA-IX positive sEVs by melanoma cells in vivo, plasma samples from a pool of five late stage BRAFV600E-positive melanoma patients were processed using an anti-CA-IX-immuno-isolation method and a commercially available kit for generic EV precipitation. DNA was extracted from the isolated sEVs and used to detect the melanoma specific mutation BRAFV600E by qPCR. Immuno-isolation with the anti-CAIX-antibody improved the detection of the melanoma-specific BRAFV600E gene compared to the generic EV precipitation, indicating the enrichment of tumor-derived sEVs carrying the mutation (Figure 6). Consistent with these data, the generic EV precipitation protocol isolated higher levels of DNA containing the BRAF WT gene, suggesting the co-isolation of more sEVs of non-tumor origin.

## 3. Discussion

In the present study, we investigated how hypoxia influenced the release of sEVs in melanoma and focused on the characterization of sEVs from two representative in vitro melanoma cell lines, which were selected based on the study of Rossi et al. [48]. A375 cells are associated with a highly aggressive phenotype with high migratory capability and invasiveness, while SK-MEL-28 cells have an intermediate phenotype [53]. Both cell lines expressed bona fide markers of hypoxia, HIF1α, and CA-IX when grown under low oxygen conditions. Conditioned media from these cell cultures contained small extracellular vesicles (sEVs), but their concentration, median size, and size distribution did not change significantly during hypoxia. These results are not consistent with other reports indicating hypoxia as one of the factors that may increase the production and/or release of sEVs [54,55] and corroborate the idea that these mechanism are highly cell-specific.

It is known that hypoxia-induced EVs carry biomolecules involved in cancer progression, which can be exploited as potential biomarkers or as targets for therapy [56,57,58]. CA-IX is a protein overexpressed in melanoma and is associated with the aggressive behavior of different types of tumors; however, only a few studies have indicated the association of this protein to sEVs [59,60,61]. Interestingly, we found that hypoxia significantly increased the CA-IX mRNA and protein expression only in sEVs from the most aggressive melanoma cell line, A375, while the expression of this biomarker was not detectable on the SK-MEL-28-derived sEVs under the same conditions. These differences can be explained by the known CA-IX protein expression in higher aggressive tumor phenotypes. The ELISA assay with detergent treatment demonstrated that hypoxia-induced CA-IX is associated with the surface of A375 sEVs, and may suggest a role for this isoform in the pathogenesis of melanoma. Consistent with this hypothesis, selective immuno-isolation of CA-IX-positive sEVs from the plasma of melanoma patients improved the detection of melanoma-specific BRAFV600E mutation compared to EVs isolated with a generic precipitation method.

## 4. Materials and Methods

### 4.1. Cell Culture

Two different melanoma cell lines, SK-MEL-28 and A375, were provided by Dr. Francesca Chiarini (University of Bologna, Bologna, Italy) and Prof. Luisa Bracci (University of Siena, Bologna, Italy), respectively. The A375 cell line is characterized by a more aggressive phenotype compared to the SK-MEL-28 cell line. Cells were grown in RPMI 1640 or DMEM 10% fetal bovine serum (FBS) with antibiotics and L-glutamine added (Euroclone, Devon, UK). SK-MEL-28 and A375 cells were maintained in a humidified atmosphere at 37 °C, 5% CO_2_, 20% O_2_ or 2% O_2_, which represents a normoxic or hypoxic environment. At least 200 mL of supernatant of each cell line was collected for the isolation of sEVs.

### 4.2. SEV Purification

For each condition, 200 mL of Cell Conditioned Medium (CCM) was pre-cleared by differential centrifugation at 300× *g* for 10 min, 1200× *g* for 20 min, and 10,000× *g* for 30 min at 4 °C. Pre-cleared CCM was concentrated by ultrafiltration with an Amicon^®^ Stirred Cell with Ultracel 100 kDa Ultrafiltration Discs (Merck Millipore, Burlington, MA, USA) to a final volume of 10 mL. Concentrated samples were purified by size-exclusion chromatography (qEV10 35 nm, Izon Science, Christchurch, New Zealand). Columns were pre-equilibrated with 1X 0.22 μm filtered PBS, 10 mL of concentrated CCM was loaded in column, then 1X 0.22 μm filtered PBS was used as the mobile phase. The eluate was immediately collected in 60 fractions of 1 mL each. All of the fractions were characterized by the micro BCA assay (Thermo Fisher Scientific, Cleveland, OH, USA) and by the CD9 sandwich ELISA assay (Exosomics, Siena, Italy). CD9 positive fractions with a low protein concentration were pooled and concentrated to 0.5 mL by 100 kDa ultrafiltration (Amicon^®^ Ultra-15 Centrifugal Filter Unit, Merck Millipore, Burlington, MA, USA). Purified EVs were aliquoted and stored at −80 °C.

### 4.3. SEV Characterization: Nano Flow Cytometry

To measure the EV size and particle concentration, we analyzed samples using a Flow NanoAnalyzer (nanoFCM Inc., Nottingham, UK), which enables single particle analysis in sheathed flow. The instrument was aligned and calibrated with size and concentration standard beads (nanoFCM Inc., Nottingham, UK), which allowed for the characterization of EVs between 40 nm and 200 nm. The samples and blank (1X PBS) were read at a constant pressure of 1 kPa for 2 min and at a maximum event rate of 12,000 events/min. Between samples, the instrument was cleaned with 1X cleaning solution (nanoFCM Inc., Nottingham, UK) and the capillary rinsed with HPLC-grade water.

### 4.4. RNA Extraction

RNA was extracted from 6 × 10^9^ total particles for each condition. A total of 700 μL of TRIzol™ (Thermo Fisher Scientific, Cleveland, OH, USA) was added to each sample. Samples were vortexed for 30 s and incubated for 5 min at RT. Then, 140 μL of pure chloroform was added, the tubes were shaken for 30 s and incubated for 10 min at RT, then 1 min in ice. Phases were separated by centrifugation for 10 min at 12,000× g at 4 °C and the aqueous phase was transferred to a clean tube. Pure ethanol was added to each sample (2:1) and tubes were inverted 4–5 times. The mixture was transferred into a RNA spin column (Epoch Life Science, Missouri City, TX, USA) and centrifuged for 30 s at 14,000× g at RT, discarding the flow-through. Columns were washed 3 times with 400 µL of Wash Buffer (165 mM NaCl, 33 mM Tris HCl, 3.3 mM EDTA in MilliQ water, pH 7.4–7.5), centrifuging and discarding the flow-through as described above. RNA was eluted with 15 µL of nuclease-free water.

### 4.5. Droplet Digital PCR

Gene expression ddPCR reactions were performed in a duplex configuration using commercially available assays to amplify the beta actin and CA-IX targets (Assay IDs: dHsaCPE5190200; dHsaCPE5055974; Bio-Rad, Hercules, CA, USA) containing HEX and FAM-conjugated reporter probes, respectively. For each sample, a 20 μL one-step ddPCR reaction was prepared (One-Step RT-ddPCR Advanced Kit for Probes, Bio-Rad, Hercules, CA, USA), diluting each assay 1:20 and adding 5 μL of RNA sample. Droplets were generated with a QX200™ Droplet Generator (Bio-Rad, Hercules, CA, USA) and transferred to 96-well PCR plates. Amplification conditions were set as follows. Reverse transcription: 42 °C for 60 min; enzyme activation: 95 °C for 10 min; amplification: 39 cycles of denaturation at 95 °C for 30 s and extension at 55 °C for 1 min, with a ramp rate of 3 °C/s; enzyme inactivation: 98 °C for 10 min. Droplets were read in a QX200 Droplet Reader (Bio-Rad, Hercules, CA, USA) and the gene expression data analyzed with QuantaSoft™ Version 1.7 (Bio-Rad, Hercules, CA, USA).

### 4.6. Western Blot

Both cell lines were seeded at a density of 3 × 10^5^ cells/well and maintained in normoxia or hypoxia. Cells were lysed with Laemli buffer with protease inhibitors (Sigma Aldrich, Saint Louis, MO, USA), and after sonication, the total protein amount was determined with the micro BCA Protein Assay Reagent Kit (Thermo Fisher Scientific, Cleveland, OH, USA). A total of 30 μg of protein for each condition was loaded on 10% acrylamide gel. The SEV samples were prepared mixing 2 × 10^9^ total particles with Laemmli sample buffer 4X (Bio-Rad, Hercules, CA, USA) containing beta-mercaptoethanol and incubated at 95 °C for 10 min. Samples were loaded on precast 4–20% Mini-PROTEAN^®^ TGX Gel, 12-well, 20 µL per well (Bio-Rad, Hercules, CA, USA), then the proteins were transferred onto a 0.2 µm nitrocellulose membrane using the Trans-Blot Turbo Transfer System with the Trans-Blot Turbo Transfer Pack (Bio-Rad, Hercules, CA, USA). Western blotting were performed using EveryBlot Blocking Buffer (Bio-Rad, Hercules, CA, USA) for the blocking step and to dilute the primary antibodies and the secondary antibodies. Primary antibodies against HIF-1α (BD Biosciences, San Jose, CA, USA, 1:200 Cat. no. 610958), CA-IX (Cell Signaling, Danvers, MA, USA, 1:500 Cat. no. 5649S), beta actin (Sigma-Aldrich, Saint Louis, MO, USA, 1:50,000 Cat. no. A3854), and Alix (Santa Cruz, Dallas, TX, USA, 1:500, Cat. no. sc-271975) were used. Chemiluminescence was produced using the SuperSignal™ West Femto Maximum Sensitivity Substrate (Thermo Fisher Scientific, Cleveland, OH, USA).

### 4.7. ELISA Assay for CA-IX-Positive sEVs

A specific sandwich ELISA assay was used in order to demonstrate that the CA-IX protein is associated with EV membranes. ELISA plates, coated with a capture antibody against CA-IX, were provided by Exosomics SpA. Equal amounts of samples were added to the wells (1 × 10^9^ total particles). To lyse the EV membrane, the same number of particles for each condition was treated with RIPA buffer, containing the protease inhibitor (Thermo Fisher Scientific, Cleveland, OH, USA) and loaded onto the plate [62]. OD was read at 450 nm using a microplate reader (CLARIOStar Plus, BMG Labtech, Ortenberg, Germany).

### 4.8. Biological Samples and Patient Consent

Plasma samples from patients with unresectable metastatic melanoma stage IIIC–IV were collected for this study from January 2011 to December 2014 and followed-up until December 2016. The study was approved by the Ethical Committee of the Azienda Ospedaliera Universitaria Le Molinette, Torino, Italy and plasma samples were collected from patients after informed consent (Comitato Etico Interaziendale Titolare A/2.10 del 03-07-2014 Prot. N. 0068188). To obtain plasma for the study, 10 mL of blood was collected in K2-EDTA Vacutainer^®^ tubes (Becton Dickinson, Franklin Lakes, NJ, USA, Cat. no. 366643) using a butterfly system with a needle gauge of 21 (Becton Dickinson, Franklin Lakes, NJ, USA, Cat. no. 367281). Blood tubes were then mixed 8 times and processed within 4 h from harvest by centrifugation at 1500× *g* for 15 min at RT. Plasma was aliquoted in labeled cryotubes and stored at −80 °C prior to use. Before EV isolation, all plasma samples were centrifuged at 1200× *g* for 20 min at 10 °C to eliminate red blood cells and cellular debris.

### 4.9. Isolation of CA-IX-Positive sEV Population and Generic EV Population from Plasma Samples

An anti-CA-IX antibody and an isotype control antibody (Exosomics SpA, Siena, Italy) were used for the selective immuno-capture of CA-IX-positive sEVs, and for the capture of a generic population of EVs, respectively, from the plasma of melanoma patients. Latex beads (Magsphere, Pasadena, CA, USA) was coated using 1 µg of antibody per sample. Then, 10 µL of antibody-coated beads was added to the pre-cleared diluted plasma sample (1:1 with 1X PBS). Samples were mixed by pipetting and incubated for 2 h at RT under rotation. After incubation, samples were centrifuged at 5000× *g* for 10 min at RT. The supernatants were carefully discarded without disturbing the bead pellets. The pellets were washed two times with 1X PBS and spun down by centrifugation at 5000× *g* for 10 min. The final pellets were resuspended in 200 μL of 1X PBS. To isolate a more generic EV population and purify the associated DNA, plasma samples were processed with a commercially available kit following the manufacturer’s instructions (SeleCTEV: Tumor DNA Enrichment Kit, HansaBioMed, Tallinn, Estonia). The same volumes of pre-cleared diluted plasma samples (1–2 mL) were used for immuno-capture and generic EV isolation.

### 4.10. DNA Extraction from Isolated sEVs

Bead-bound sEVs were treated with the reagents included in the SeleCTEV: Tumor DNA Enrichment Kit (HansaBioMed, Tallinn, Estonia). Bead pellets were lysed with lysis buffer and digested with proteinase K to release the DNA from protein complexes. The samples were then supplemented with pure ethanol, loaded onto silica membrane spin columns, and centrifuged at 10,000× *g* for 1 min. Following centrifugation, the flow-through was discarded. Two washing steps were performed according to the manufacturer’s instructions to remove the contaminating solvents and plasma-derived inhibitors. Purified sEV-DNA was eluted in a final volume of 15 µL of elution buffer.

### 4.11. PCR Amplification of BRAF Gene from EV-DNA

PCR amplification of EV-DNA is challenging due to the low abundance and high fragmentation of the template. Therefore, a pre-amplification step was included in the protocol upstream of the quantitative real-time PCR (qPCR) to improve the detection of the BRAF gene.

The following pre-amplification primers and probes were used:(a)BRAF WT FW: 5′-TAGGTGATTTTGGTCTAGCTACAG+T-3′;(b)BRAF WT RW: 5′-TTAATCAGTGGAAAAATAGCCTCA-3′;(c)BRAF V600E FW: 5′-TAGGTGATTTTGGTCTAGCTACAG+A-3′;(d)BRAF V600E RW: 5′-TTAATCAGTGGAAAAATAGCCTCA-3′.

The following qPCR primers and probe were used:(a)WT: FW: 5′-TAGGTGATTTTGGTCTAGCTACAG+T-3′;(b)WT RW: 5′-TTAATCAGTGGAAAAATAGCCTCA-3′;(c)V600E FW: 5′-TAGGTGATTTTGGTCTAGCTACAG+A-3′;(d)V600E RW: 5′-TTAATCAGTGGAAAAATAGCCTCA-3′.(e)Probe: 5′-FAM-CCGAAGGGGATC+CAGACAA+CTGTTCAAACTGCCTTCGG-3BHQ1-3

All reagents were thawed at RT for at least one hour and briefly mixed without vortexing to avoid inactivation of the enzyme. Each pre-amplification reaction included 7 µL of eluted DNA, 1X Bioron High Fidelity Buffer, 3 mM MgCl_2_; 200 µM dNTPs, 1.25 units of SNPase polymerase (Bioron GmbH, Römerberg, Germany), and 0.4 µL of primers (10 µM) in a total volume of 20 µL. Each reaction was performed in a PCR-compatible microvial loaded onto a thermal PCR cycler running the following PCR program: 98 °C for 30 s, 98 °C for 10 s, and 72 °C for 5 min, 4 °C on hold. The pre-amplified DNA was diluted in 80 µL of nuclease-free water and immediately used for qPCR analysis. For the amplification of target genes from EV-DNA, each qPCR reaction included 7 µL of pre-amplified DNA, 1X SsoAdvanced Universal Probes Master mix (Biorad, Hercules, CA, USA), 0.625 µL of primers (10 µM), and 0.3125 µL of fluorescent probe (10 µM) in a total volume of 25 µL. After careful mixing, each reaction was loaded in duplicate on a 96-well PCR plate and the following qPCR program was launched: 95 °C for 3 min, 40 cycles at 95 °C for 5 s, and 60 °C for 30 s, followed by a final hold step at 4 °C.

### 4.12. Statistical Analysis

A two-tailed Student’s *t*-test was applied to evaluate the statistical difference between the two isolation methods from the plasma samples. Statistical comparison that output *p*-values < 0.05 were deemed significant.

### 4.13. Graphical Abstract

Graphical abstract figure was created with BioRender.com.

## 5. Conclusions

Small EVs (sEVs) are found in biofluids and protect biomolecules that can be exploited as potential biomarkers for cancer monitoring, diagnosis, or screening. In this study, we characterized sEVs released from melanoma cells, exposed to hypoxia, and confirmed that the hypoxia-biomarker CA-IX is associated with the membrane of melanoma-derived sEVs under this condition. CA-IX-positive sEVs were purified from the plasma of metastatic melanoma patients, and interestingly enriched the melanoma specific mutation BRAFV600E. These results warrant further investigation but suggest the possibility of exploiting EV-associated CA-IX for liquid biopsy applications and cancer diagnosis.

## Figures and Tables

**Figure 1 ijms-24-06122-f001:**
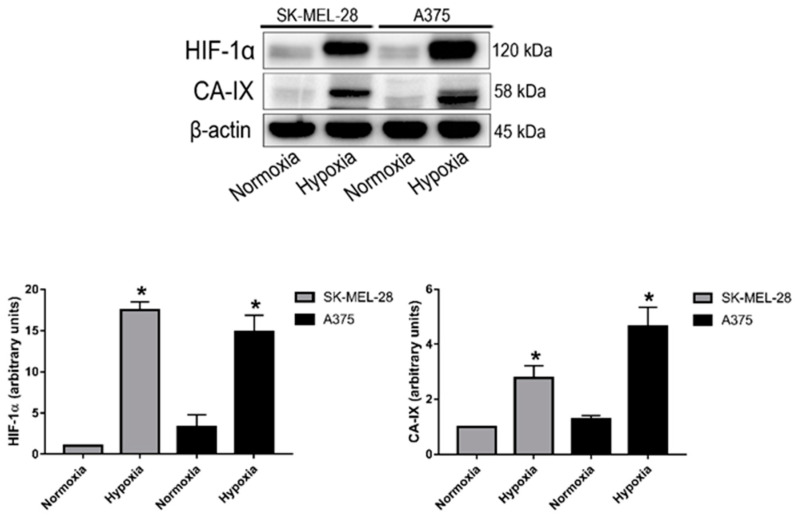
HIF-1α and CA-IX protein expression in the A375 and SK-MEL-28 cell lines after 24 h exposure to normoxia or hypoxia. Blots are representative of two independent experiments and beta actin was used as the loading control (* *p* < 0,05 indicates a statistically significant difference).

**Figure 2 ijms-24-06122-f002:**
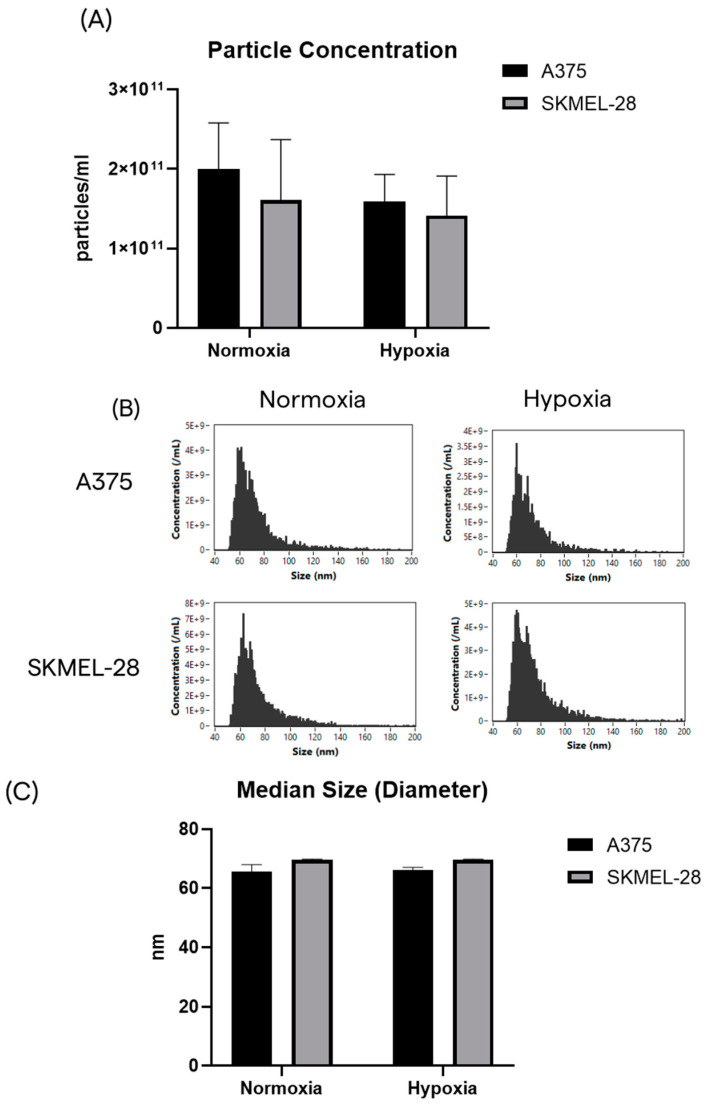
The A375 and SK-MEL-28 cells release similar populations of sEVs in terms of number and size. (**A**) The particle yield after SEC chromatography, starting from the same volume of CCM for each condition. (**B**) Size distribution profiles by nFCM. (**C**) Median size of purified sEVs from the A375 and SK-MEL-28 cell lines.

**Figure 3 ijms-24-06122-f003:**
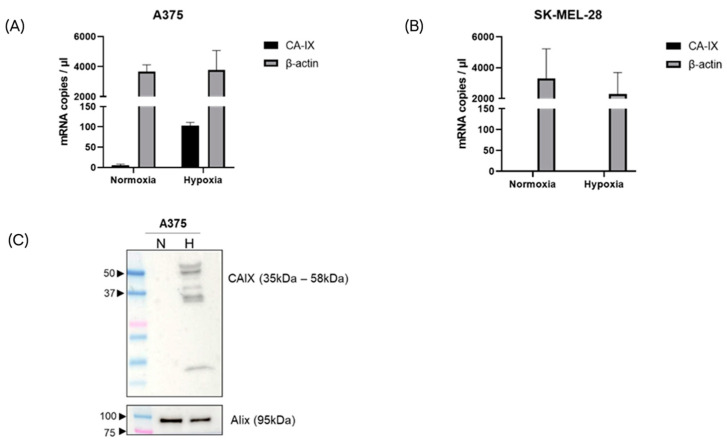
CA-IX mRNA expression in melanoma-derived sEVs, released under normoxic and hypoxic conditions. After RNA isolation, CA-IX and beta actin mRNA copies were measured by ddPCR in the A375 model (**A**) and SK-MEL-28 model (**B**). (**C**) CA-IX protein expression was induced in A375-derived sEVs released under hypoxia. Alix blot was used to confirm the EV identity.

**Figure 4 ijms-24-06122-f004:**
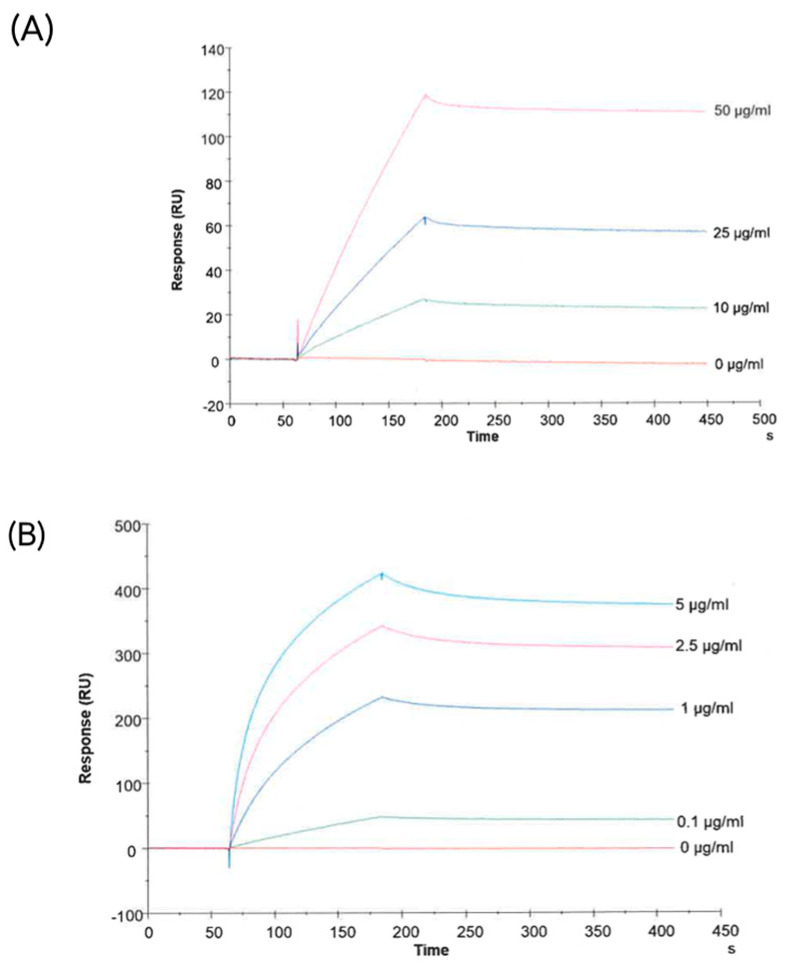
Characterization of an antibody against human CA-IX. Figures show the binding of the antibody with CA-IX-positive sEVs (from HEK293pRTS-CA9 cells) (**A**) and with the recombinant protein (**B**).

**Figure 5 ijms-24-06122-f005:**
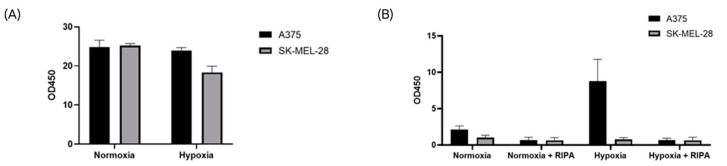
The CA-IX protein is associated with the membrane of hypoxic sEVs. EV identity was confirmed for all samples with the CD9 ELISA assay (**A**). Only A375 cells released CAIX-positive sEVs under hypoxia, and the CAI-X protein was associated with the EV membrane (**B**).

**Figure 6 ijms-24-06122-f006:**
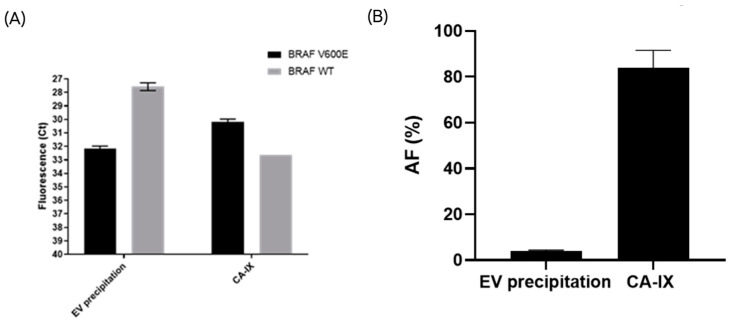
The CA-IX-based immune-isolation of sEVs from metastatic melanoma liquid biopsies improved the detection of the BRAFV600E mutation compared to the generic EV precipitation method. Data are expressed as the threshold cycle (Ct) values and plotted on an inverted Y scale (**A**). Mutant BRAFV600 allelic frequency (%) was calculated for the two isolation methods. A statistically significant difference was found between the two methods (*p*-value = 0.04) (**B**).

## Data Availability

The data presented in this study are available on request from the corresponding author.

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
