# Peer review of "CA-IX-Expressing Small Extracellular Vesicles (sEVs) Are Released by Melanoma Cells under Hypoxia and in the Blood of Advanced Melanoma Patients"

_ijms, 2023, doi:10.3390/ijms24076122_

Round 1
Reviewer 1 Report
The manuscript "CA-IX-expressing small extracellular vesicles (sEVs) are re- 2 leased by melanoma cells under hypoxia and in the blood of 3 advanced melanoma patients" shows that two melanoma cell lines release small extracellular vesicles(sEVs) in vitro. Under hypoxia conditions, one of the cell lines A375 releases sEVs containing CA-IX mRNA and protein. In the melanoma patient's plasma, enriched CA-IX-positive sEVs showed a higher copy of BRAFV600E mutation gene, suggesting CA-IX-positive sEVs may be a potential biomarker for liquid biopsy. The contents are of broad interest. With that said, some limitations raise concerns.
- The authors only test two melanoma cell lines, which weakens the conclusion, it would be better to assess more cell lines.
- Only one of two melanoma cell line release CA-IX-positive sEVs under hypoxia conditions, the author should at least discuss what makes the difference.
- For figure 4, the resolution is very low. The explanation in the text and the figure legend is not matching.
- "After detergent treatment with RIPA buffer, the positive signal from A375 hypoxic sEVs was abrogated due to the disruption of EV membranes (Figure 5B). These results indicate that CA-IX protein is expressed on the surface of A375 sEVs released under hypoxia". The results show RIPA might destroy the binding between antibody and CA-IX protein. more experiments are required for supporting the conclusion they made.
- For Fig.6, the copy of CA-IX in each group should be included to show the isolation does work. And it would be better to have a control to show that the input for RT-PCR for each group is the same.
Reviewer 2 Report
The authors have an original research scope on melanoma biology focusing on small EVs (sEVs) harbouring a unique expression of CA-IX, which is also demonstrated in the clinical samples derived from BRAF V600E-positive melanoma patients. The marker of CA-IX looks very convinsing in cell culture experiments as clearly shown in the manuscript. However, some essential details can be covered in the section of clinical study.
Figure 6.
1. I wonder if any statistics could support a significant finding on the analysis.
2. Besides CA-IX immuno-isolation of sEVs, would it be possible to quantify mRNA or protein level of CA-IX in the generally precipitated sEV samples?
3. Ct values are not sufficient enough for quantification. Instead, comparable ⊿CT or a standard-based measurement of BRAF gene mutation should be considered.
Round 2
Reviewer 1 Report
The authors addressed all my questions.